# Rapid Determination of the ‘Legal Highs’ 4-MMC and 4-MEC by Spectroelectrochemistry: Simultaneous Cyclic Voltammetry and In Situ Surface-Enhanced Raman Spectroscopy

**DOI:** 10.3390/s22010295

**Published:** 2021-12-31

**Authors:** Jerson González-Hernández, Colby Edward Ott, María Julia Arcos-Martínez, Álvaro Colina, Aránzazu Heras, Ana Lorena Alvarado-Gámez, Roberto Urcuyo, Luis E. Arroyo-Mora

**Affiliations:** 1Centro de Investigación en Electroquímica y Energía Química (CELEQ), Universidad de Costa Rica, San José 11501-2060, Costa Rica; jerson.gonzalez@ucr.ac.cr (J.G.-H.); alagamez@yahoo.com (A.L.A.-G.); Robertourcuyo@gmail.com (R.U.); 2Escuela de Química, Universidad de Costa Rica, San José 11501-2060, Costa Rica; 3Department of Forensic and Investigative Science, West Virginia University, Morgantown, WV 26506, USA; ceo0009@mix.wvu.edu; 4Departamento de Química, Universidad de Burgos, Pza. Misael Bañuelos s/n, E-09001 Burgos, Spain; jarcos@ubu.es (M.J.A.-M.); acolina@ubu.es (Á.C.); maheras@ubu.es (A.H.); 5Centro de Investigación en Ciencias e Ingeniería de Materiales (CICIMA), Universidad de Costa Rica, San José 11501-2060, Costa Rica

**Keywords:** spectroelectrochemistry, synthetic cathinones, Raman SERS, 4-MMC, 4-MEC

## Abstract

The synthetic cathinones mephedrone (4-MMC) and 4-methylethcathinone (4-MEC) are two designer drugs that represent the rise and fall effect of this drug category within the stimulants market and are still available in several countries around the world. As a result, the qualitative and quantitative determination of ‘legal highs’, and their mixtures, are of great interest. This work explores for the first time the spectroelectrochemical response of these substances by coupling cyclic voltammetry (CV) with Raman spectroscopy in a portable instrument. It was found that the stimulants exhibit a voltammetric response on a gold screen-printed electrode while the surface is simultaneously electro-activated to achieve a periodic surface-enhanced Raman spectroscopy (SERS) substrate with high reproducibility. The proposed method enables a rapid and reliable determination in which both substances can be selectively analyzed through the oxidation waves of the molecules and the characteristic bands of the electrochemical SERS (EC-SERS) spectra. The feasibility and applicability of the method were assessed in simulated seized drug samples and spiked synthetic urine. This time-resolved spectroelectrochemical technique provides a cost-effective and user-friendly tool for onsite screening of synthetic stimulants in matrices with low concentration analytes for forensic applications.

## 1. Introduction

The United Nations Office on Drugs and Crime (UNODC) estimates that 5.4% of the world’s adult population has used drugs at least once in the previous year. Even more serious, around 0.7% are affected to the point of suffering dependence or requiring treatment [1]. Considering the devastating consequences of drug misuse, abuse and addiction, the control of these drugs continues to be a recurrent problem.

In the last decade, there has been a dramatic expansion of the novel psychoactive substances (NPS) market. According to UNODC’s annual report of 2020, the number of NPS increased from 166 between 2005–2009 to 950 by the end of 2019, while the total number of psychoactive substances under schedule was 230 in 1990 and increased to 282 in 2018. These chemical compounds are commonly referred to as ‘legal highs’ because they may be bought without legal restrictions through head shops and online websites; however, they pose a growing threat to public health similar to those illegal drugs listed in the Convention on Narcotic Drugs or Psychotropic Substances [2,3].

The stimulants market appears to be continuously growing and changing. In many regions of the world, mainly in Central Asia and Eastern Europe, synthetic drugs are replacing opiates. Evidence suggests that methamphetamines and cathinones are widely available today [1]. The European Monitoring Centre for Drugs and Drug Addiction (EMCDDA) currently monitors at least 138 synthetic cathinones, an increasing group of stimulants that dominate the seizures of novel psychoactive substances in the continent [4].

Mephedrone (4-MMC) and 4-methylethcathinone (4-MEC) are synthetic cathinones that are similar to amphetamine, methamphetamine, and ecstasy in structure and mechanism of action, predominantly as central nervous system stimulants [5,6]. These designer drugs are usually encountered as research chemicals, plant food, bath salts or glass cleaner and are sold in powder, pill, or capsule forms [7]. Mephedrone and its derivates are some of the more commonly seized synthetic cathinones. Therefore, it is crucial to obtain an immediate response from analytical methods that can be applied in the field to take timely actions in seizure situations and provide investigative leads.

In an effort to quickly respond to a growing need to streamline the delivery of results, several alternative drug testing approaches have been recently suggested [8,9]. For example, electrochemical techniques such as voltammetry [10,11,12,13] and amperometry have been utilized as novel forensic tools for the detection of drugs [14,15]. These electrochemical techniques adapt well to the required field conditions [16], offering screening opportunities that can provide on-site decisions to law enforcement before sending evidence to the forensic lab for later analysis and processing [17].

In order to overcome limitations in discriminating between electro-active analytes, additional analytical tools are needed to increase the selectivity of the electrochemical examination. This is especially true when working with drugs that may have been mixed with cutting agents and chemical adulterants, and to distinguish between drugs with similar chemical structures [18]. Under these circumstances, a parallel and more confirmatory test such as Raman spectroscopy can resolve this limitation [19]. Several strategies are available to boost the Raman signal including SERS through the use of nanostructures such as nanostars [20,21], nanoparticles [22], nanopillars [23] or nanotubes [24] that induce surface-enhanced scattering [25,26], allowing the collection of spectra for molecules at low concentrations or even of individual molecules [27]. Our group has reported the behavior of Cu centers of enzymes using nanostructured Au as SERS substrates [28].

During a combined EC-SERS experiment, the metal surface activation occurs in situ in a controlled and straightforward way. The voltammetric pretreatment at the electrode surface allowed an efficient and reproducible development of roughened nanostructures [29,30,31]. The advantage of this procedure is that it combines Raman spectroscopy with electrochemical analysis, thus offering both identification and quantification capabilities [32]. Several EC-SERS miniaturized sensors have been described in a recent review, covering quality control, diagnosis, biomedical, environmental and food safety fields [33]. A novel electrochemical roughening technique of commercial disposable electrodes of metallic surfaces such as gold and silver has been used to improve the reproducibility of the SERS substrate based on the deposition of nanostructures [34,35,36,37]. Additionally, screen-printed electrodes (SPE) have been recently reported as excellent substrates for SERS experiments [38,39,40].

This manuscript presents a time-resolved spectroelectrochemical method for detecting two synthetic cathinones -4-MMC and 4-MEC- using a gold SPE electrode as SERS substrate (AuSPE). These two cathinones (Figure 1) are structurally similar, differing only by a CH2 group, which challenges their identification when they are present as mixtures in diluents or other drugs. Some of the advantages and limitations of the EC-SERS approach are described. The purpose of using the CV technique to generate onsite nanostructures and as a semi-quantitative tool is also discussed.

## 2. Materials and Methods

### 2.1. Chemicals

The Analytical Reference Standards of 4-MMC hydrochloride in the form of crystalline powder and 4-MEC hydrochloride in methanolic solution and crystalline powder were supplied by Cayman Chemical Company. The standards were reconstituted in methanol 99.99% purchased from Fisher Scientific (Optima^®^ Grade) at different concentrations. Sulfuric acid (H_2_SO_4_) was supplied by Fisher Scientific and D-(+)-maltose monohydrate by MP Biomedicals LLC. All solutions were prepared with deionized water of resistivity 18.2 MΩ-cm supplied by the system Millipore Direct-Q^®^.

### 2.2. Instrumentation for Electrochemistry and Raman Spectroscopy

The portable SPELEC Raman instrument from Metrohm DropSens was used to perform the spectroelectrochemical experiments. The compact unit combines a Raman spectrometer, a bipotentiostat/galvanostat and a LASER Class 3B (785 nm). It includes an adapter for disposable SPE and an opaque cell for Raman spectroscopy analysis. This instrument uses DropView SPELEC 1.2 interface to control the hardware and analyze the data obtained.

Confirmatory voltammetric measurements were carried out using a potentiostat Autolab PGSTAT Echo Chemie 128 N (Utrecht, The Netherlands) controlled by the electrochemistry software Nova v2.1.4.

### 2.3. Substrate Characterization

Characterization of the gold electrode surface before and after in situ SERS generation was achieved using scanning electron microscopy (SEM) and atomic force microscopy (AFM). Analysis was carried out on a new electrode (pristine electrode) and after in situ electro-activation of the substrate through a positive sweep from +0.6 to +1.4 V and then returning to +0.1 V. To this end, a JEOL SEM JSM7600 operating at 5.0 keV was used to obtain images of the surface at magnification from 40× to 30,000×. A Denton Desk V Sputter coater with Carbon Rod Accessory for carbon evaporation was used to lightly coat the samples for SEM analysis.

Surface morphology of the AuSPE was studied using an Asylum Research MFP-3D^TM^ AFM. WSXM free software was used for AFM data processing [41].

The SEM images were processed and analyzed using ImageJ. The scale was set to define the pixels in terms of SEM scale. A fast Fourier transformation (FFT) and a band-pass filter were used. The threshold was adjusted to analyze particles of size larger than 0.1 μm^2^.

### 2.4. Spectroelectrochemical Determinations

The chemical standards in methanol were evaporated using a continuous flow of nitrogen from Matheson Tri-Gas, Inc. and reconstituted in H_2_SO_4_ 0.01 M. Screen-printed electrodes (EC-SERS platform) were purchased from DropSens (220BT cured at low temperature). The SPE system included a gold working electrode, a silver pseudo-reference electrode (AgSPE) and a gold counter electrode. These electrodes were selected to evaluate the reproducibility of the SERS substrate.

Raman detection was performed using an infrared 785 nm laser wavelength. The Raman probe focus was optimized using O-rings as spacers (0.5–1.5 mm) in the cell to adjust the focal distance. The laser power was optimized between 36.3 and 550.5 mW and integration time between 100 and 5000 ms. Cyclic voltammetry screening was carried out at the best instrumental conditions: the potentials were stepped in 2 mV increments from +0.1 to +1.2 or +1.4 V starting at 0.6 or 1.0 V and a scan rate of 50 mV/s. The electrochemical measurements and Raman spectroscopy were performed simultaneously on a drop of 50–70 μL solution pipetted on the electrode placed into the cell.

### 2.5. Raman Measurements

Raman data in the solid-state were acquired using the SPELEC instrument in the mode for Raman spectroscopy only using a spacer of 0.5 mm. A small amount of each sample was deposited onto aluminum foil placed inside the Raman cell forming a small powder film. The spectra were recorded with an integration time of 10 s and a laser power of 464.8 mW.

### 2.6. Interference Study

For the interference effect assessment, simulated seized drug samples were prepared from solid powdered drugs and cutting agents (maltose and lidocaine). Preparation of mixtures was performed by weight to provide a ratio of drug to cutting agent 1:4. Target drug and maltose or lidocaine were mixed into plastic baggies. The samples were analyzed by placing approximately 1 mg of the seized mixture sample into a microcentrifuge tube and dissolving in 1000 µL of 0.01 M sulfuric acid. Analysis occurred in the following order to obtain EC SERS data (all CV and Raman parameters were set as detailed previously, laser power of 379.1 mW and integration time of 5000 ms):

1.0→1.4→0.1→1.0 V

0.6→1.4→0.1→0.6 V (using the same drop on the electrode)

This procedure was performed for both 4-MMC and 4-MEC simulated sample detection.

### 2.7. Reproducibility of the EC-SERS Substrate

Simulated seized samples of 4-MMC and 4-MEC cut with maltose in a ratio of solid drug to cutting agent 1:4 was tested by the simultaneous technique. The measurements were performed in triplicate on different electrode sensors to assess the Raman intensity of the two principal bands in the EC-SERS spectrum (Section 3.3).

### 2.8. Urine Analysis

Synthetic urine (Ricca Chemical Company, Arlington, TX, USA) was selected as a biological specimen to expand the scope of applicability of the method. Drug stocks were prepared at 100 μg/mL from standard chemical solutions by drying down the required amount and reconstituting in 0.1 M KCl ACS reagent ≥99% purchased from Sigma-Aldrich. Two milliliters of synthetic urine was spiked using 20 μL of the drug stocks for a final cathinone concentration of 1.0 μg/mL.

The spiked samples and blank were placed into separate 15 mL conical tubes. An aliquot of 400 µL of 10% ammonium hydroxide ACS Plus grade from Fisher Scientific was added to raise the pH to approximately 11.5. The tubes were vortexed for 30 s before and after adding 2.0 mL of methyl tert-butyl ether (MTBE) HPLC grade supplied by Fisher Scientific. They were centrifuged for one minute to ensure phase separation. The top organic layer was removed into a separate microcentrifuge tube and dried down under nitrogen flow. Finally, the chemical residues were reconstituted in 50 μL of 0.01 M sulfuric acid.

Spectroelectrochemical analyses were performed in a homemade black box prepared for Raman and conditions were set as described in Section 2.4 and Section 2.6.

## 3. Results and Discussion

The two synthetic stimulant drugs studied are classified as secondary amines with a calculated pKa around 8.1 [42]. The electrochemical potential of these weak bases is influenced by the pH of the medium in which the measurement is performed. Sulfuric acid at a pH of 1.8 made it possible to work with a wider electrochemical window when gold electrodes were used [43]. These conditions enabled the separation of the two oxidation waves corresponding to the analyte and the substrate.

### 3.1. Determination of the Electro-Activity

The electro-activity of the target drugs was determined via cyclic voltammetry (CV). The experiment was conducted in the positive direction, starting the sweep at +0.60 V, which allowed the resolution of the peaks to the gold’s oxidation wave used as the working electrode. The electrochemical process of 4-MMC and 4-MEC using AuSPE is outlined in Figure 1, where the average of the three measurements for each concentration is plotted. The electro-oxidation of both substances occurs at a potential of around +0.91 V (peak I) as suggested by the growth of the current peak when increasing the concentration from 50 to 100 μg/mL, while gold oxidizes around +1.1 V (peak II). The prominent cathodic peak at +0.62 V for drug samples (peak III) or +0.52 V for the blank, corresponds to the reduction of the gold compounds previously formed in the positive scan. Both oxidation and reduction peaks are shifted towards less positive potentials for the blank. The voltammograms demonstrate the common hysteresis of the oxide formation–reduction behavior that some metals such as gold undergo in the electron transfer reactions [44].

In the spectroelectrochemical analysis, voltammetric scanning has the dual role of tentatively determining the amount of analyte in the sample while performing in situ pretreatment on the electrode surface, which serves as a substrate for the acquisition of Raman spectra. Attention is now turned toward the quantitative application of this technique (see Figure 2). Calibration curves of peak height vs. analyte concentration were obtained using cyclic voltammetry (CV) on the AuSPE. The generation of these curves opened the possibility to utilize the method for quantitative purposes, which is desirable in forensic and other analytical fields. One of the limitations of using CV is the overall low sensitivity offered mainly due to the susceptibility to residual currents encountered [45]. Therefore, at low concentrations, the shape of the analyte peaks is more difficult to recognized from the gold oxidation peak as shown in Figure 2B. The corresponding calibration plots show a linear response: $I_{p}=0.064\cdot C-\;0.532$ for 4-MMC (Figure 2A) and $I_{p}=0.119\cdot C-\;3.92$ for 4-MEC (Figure 2B) with regression coefficients (*R*^2^) of 0.997 and 0.999, respectively. The limit of detection (LOD) was estimated at three times the standard deviation of the linear regression divided by the slope of the linear curve (3σ/S). The method for 4-MMC exhibited a LOD of 6.6 and 2.4 μg/mL for 4-MEC. Finally, time-resolved electrochemical and spectroscopic information provides the necessary methodology to perform the selective determination of these two designer drugs.

### 3.2. Substrate Characterization

The dynamic electrochemical process in the SPE allows the surface of the gold to become SERS-active by roughening the metal surface through the oxide formation-removal reaction. The substrates were characterized with SEM images, as shown in Figure 3. Au microparticles form the surface of the pristine SPE; however, this rough surface with some cavities is not sufficient to induce the SERS response, as will be shown below. In Figure 3A,B, a slight alteration is noted in the morphology of the electro-reduced substrate to the unaltered electrode. Au nanoparticles are electrochemically deposited on the surface of the Au microparticles, obtaining a high density of the nanoparticle aggregates [46]. The particle-size distribution analysis demonstrated an increase of 17% for nanoparticles with a size of 0.1 μm and 50% for nanoparticles with a size around 0.3 μm after voltammetric treatment. If the images are analyzed at a higher magnification ratio (Figure 3C,D), it is observed that the nanoparticles of the in situ electro-activated substrate are aggregated, yielding a homogeneous distribution of metallic particles that are rougher and with greater surface area, increasing the probability of generating the excitation of the surface plasmon [47].

In the SERS mechanism, the long-range effect produces a much stronger electric field close to the substrate surface, which increases the relative SERS intensity. According to the short-range effect, the polarizability tensor of the ligand is perturbed by chemical bond formation with the metal or by charge-transfer, giving a redistribution of the electron density between the molecular states and the energy levels of the metal conduction band [48]. These two effects result in improved resolution of the spectra for the identification of the molecule. Nonetheless, SERS electromagnetic theory requires that the size of the metal particles be much smaller than the wavelength of the exciting radiation (Rayleigh approximation). The primary enhancement derives from the resonance between the incident radiation and the electronic excitation wave on the metal surface, called the surface plasmon band [49]. As such, a face roughness on a nanometer length scale is necessary. On the electrode surface, these conditions are obtained after in situ electro-deposition of AuNPs aggregates. The 3D AFM image of the pristine AuSPE substrate (Figure 4A) shows a microscopic roughness, as well as deep and micrometer-scale cavities. In contrast, a smoother and restructured surface is observed in Figure 4B. This electrochemical modification may change the size cavities to the desirable nanometer range and consequently promote SERS phenomena.

### 3.3. Spectroelectrochemical Sensing

Figure 5 depicts the voltammetric profile and the EC-SERS effect of a diluted 4-MMC solution on the AuSPE in a simultaneous measurement. The potential-dependent evolution of spectra—acquired at potentials marked with an asterisk—shows an active surface just after the electro-reduction of the gold oxides. This critical change suggests an increase in the adsorption of the target molecules on the electro-synthesized AuNPs and generates the overall SERS effect necessary to identify some characteristic bands.

The EC-SERS spectrum, which is shown in an orange color (Figure 5B), appears at 0.45 V in the negative direction when the substrate is chemically activated by electrochemical deposition of gold nanoparticles. It is noteworthy that neither the initial Raman spectrum at 1.0 V in the positive direction nor the spectra at 1.35 and 0.75 V during the cathodic scan show any band corresponding to the molecule, confirming that the pristine SPE does not demonstrate SERS properties. Raman spectra obtained at 1.0, 1.35 and 0.75 V can be related to the ink used in the fabrication of the SPE.

The vibrational spectral analysis is based on the most significant bands: the signal arising at 804 cm^−1^ is assigned to the p-disubstituted ring vibrational mode γ(CH) out-of-plane, the 975 cm^−1^ mode may be attributed to methyl rocking vibration ρ(CH_3_) coupled with ν(CN) motion, the wavenumber at 1213 cm^−1^ results from aromatic δ(CH) in-plane deformation vibrations, and the ring stretching mode ν(CC) gives rise to the strong peak at 1605 cm^−1^ [50,51,52].

Under the spectroelectrochemical conditions for both molecules individually tested, the electro-oxidation potentials of +0.88 V for 4-MMC and +0.90 V for 4-MEC were obtained at 100 μg/mL, and the relative standard deviation (RSD, *n* = 5) was estimated at 1.0% and 2.3%, respectively. When comparing these values to the oxidation potentials previously recorded in the individual drug analysis, it was observed that in the data collected from the SPELEC instrument, there was minimal differentiation between the oxidation peaks of the two analytes. Furthermore, the peaks were slightly shifted towards lower potentials than the Autolab results.

Due to the structural similarity of both molecules (Figure 1), these substances share a few physical and chemical properties that make it difficult to distinguish them by electrochemical techniques as a rapid screening because the peaks become wider as the concentration increases, causing slight shifts in the potential at which the current reaches a maximum (seen in Figure 6A).

Electrochemical methods are useful for quantification because the characteristics of the oxidation wave depend on the concentration of the analyte; however, for a rapid and accurate identification, it is essential to use a more selective analytical tool that allows for the unequivocal characterization of the substance in solution [53,54]. Despite the mentioned similarity of the drugs studied, it is possible to discriminate the chemical structures accurately by EC-SERS in a quick and simple analysis that can even be applied in the field due to the portability of the SPELEC instrument. Figure 6B depicts EC-SERS spectra of both substances in acid solution. The most significant difference between the two spectra is in the region of the gray dashed rectangle (1150–1225 cm^−1^). The spectrum of 4-MMC shows a triplet with signals at 1161, 1185 and 1213 cm^−1^, which could be attributed to aromatic δ(CH) in-plane deformation vibrations [55,56], while for 4-MEC, only one doublet is distinguished with the signals at 1185 and 1213 cm^−1^. The absence of the band at 1161 cm^−1^ in the 4-MEC EC-SERS spectrum agrees with the results in Figure 6C for the spectra acquired from substances in the form of crystalline powder, allowing for the correct identification of each of the stimulants studied, as it has been reported [57,58].

The advantage of activating an optimal surface morphology to induce the SERS effect is its ability to analyze a low concentration of the analyte, according to the results. In comparison, these low concentrations may not be detected using non-SERS Raman due to a decrease in sensitivity, and therefore a loss in peak resolution compared to SERS. The possibility to identify and quantify low concentrations of the target drugs gives the technique a potential use for testing both seizure samples and biological matrices [59] or even in instances where the fluorescence overwhelms the Raman signals [60]. Furthermore, in situ SERS substrate activation provides some benefits to overcoming a possible time-dependent decrease of the surface and plasmonic properties, which could affect the reproducibility and reliability of the measurements [29].

### 3.4. Interference Study

The interference effect was evaluated by testing simulated seized drugs cut with maltose and lidocaine, common cutting agents for this type of psychoactive substance [61]. The sensing of the seized sample, prepared at a ratio of drug to maltose or lidocaine 1:4, provides sufficient information to determine the presence or absence of the stimulants in the analyzed powder mixture. In this instance, the maltose, used as a cutting agent, is not electro-active under the test conditions; therefore, the oxidation signals correspond to synthetic cathinones (Figure 7A). This electrochemical technique (CV) may be applied to quantify the analyte in the mixture and determine the appropriate ratio. Additionally, the in situ EC-SERS spectrum allows the identification of drugs through the most relevant signals, which were marked in Figure 7B with dashed lines. This highlights one of the advantages of this method: the confirmation of the presence of the molecule by two techniques simultaneously in a single run. Although maltose represents 80 percent of the mixture, the characteristic bands of the drugs are prominent and easy to identify.

Regarding the synthetic cathinones cut with lidocaine, the electrochemical response is different because this cutting agent is electro-active as can be seen in Figure 8A, in which only the oxidation waves of lidocaine are distinguished under optimized working conditions for 4-MMC/lidocaine (the same results were observed for 4-MEC/lidocaine). Despite this substance does not allow the quantification of drugs by the CV technique, the simulated samples EC-SERS spectra show significant differences with respect to the spectrum of lidocaine dissolved in 0.01 M H_2_SO_4_. The identification of some characteristic bands of the stimulants, mainly the Raman shift at 804 and 1213 cm^−1^ (Figure 8B, the dashed lines indicate the position of the drug bands in the spectra of cathinones solution) is evidence that the proposed method may be a useful screening tool in preliminary tests for drug samples cut with different substances, whether solid or in solution, even at low concentrations. These previous results demonstrate the potential on-site application of this method to investigate other common mixtures in seized drug samples by a rapid procedure.

The selectivity of the synchronous experiment can be evaluated independently. The electrochemical activity of the substances present in the mixture and the oxidation waves developed at the gold electrode caused limited selectivity on the target analyte peaks. When the EC-SERS was set, the selectivity improved. Several characteristic bands of the target molecules were enhanced and allowed the proper identification of each drug with contrasting responses when the mixture involved maltose and lidocaine. Adequate identification and quantification were possible in the seized drug cut with maltose only.

The EC-SERS substrate reproducibility was assessed by measuring drug samples cut with maltose. Here, high selectivity is required to improve the identification of studied analytes. The two main bands in the corrected EC-SERS spectrum by baseline subtraction were considered to determine the Raman intensity RSD (*n* = 3). Table 1 shows a low RSD for the most intense and significant band of 4-MMC and 4-MEC at 1605 cm^−1^ (<2%). For the second band at 1213 cm^−1^, RSD values are a little higher, probably because the signal intensity is lower and the interaction and overlay with the cutting agent bands are more likely. These preliminary determinations demonstrate a good reproducibility for the SERS platform and the values for this performance parameter are sufficient for this study in which the EC-SERS technique is used as an identification tool.

### 3.5. Applicability of the Method

In order to demonstrate the feasibility and applicability of the method, synthetic urine samples were spiked with the target drugs and analyzed. Based on the possible concentration of free-mephedrone excreted in urine [62], synthetic urine samples were prepared at 1.0 μg/mL of 4-MMC and 4-MEC for testing. The drugs were extracted and reconstituted in H_2_SO_4_ 0.01 M. In the voltammetric profile of the cathinones an oxidation peak is observed that overlaps with the oxidation wave of gold required for the in situ EC-SERS effect (Figure 9A). If the goal goes beyond identification in biological samples, other more sensitive electrochemical techniques are recommended such as SWV or DPV. The characteristic bands of the analytes can be identified in the EC-SERS spectra (Figure 9B). The spectroelectrochemical data analysis demonstrates that it is possible to identify, quantify and even differentiate between 4-MMC and 4-MEC at low concentration in urine as described above, considering the discriminatory region in the spectrum from 1150 to 1225 cm^−1^.

An outstanding advantage of this method is the implementation of the SERS substrate electro-activation procedure for the quantification of the two drugs studied, either in samples mixed with their cutting agents or in biological matrices in which the analyte concentrations are low. The spectroelectrochemical approach of this work is based on the preliminary measurement of either 4-MMC or 4-MEC in its pure form or mixed with cutting agents. However, the data obtained from the same gold sensor demonstrates the feasibility of the coupled technique to be able to identify both drugs in a mixture. Efforts in this direction are important because the new generation of ‘legal highs’ are generally sold in combined with other controlled substances [63].

## 4. Conclusions

In the present work, the spectroelectrochemical sensing and comparison of the results for 4-MMC and 4-MEC were explored for the first time using a portable instrument. Both drugs were found to be electro-active on a gold electrode at pH 1.8 by the CV technique. This electrochemical sweep allows a simultaneous in situ activation of the SPE surface to induce the SERS effect. The CV vs. SERS spectroelectrochemical process enables a rapid and reliable analysis technique in which both synthetic cathinones can be selectively analyzed or detected through the characteristic bands of the EC-SERS spectrum that provide a real fingerprint of the molecule, even for molecules as similar as those studied in this work.

The time-resolved technique proposed in this study, in addition to being cost-effective and user-friendly, has a potential quantitative on-site application for samples from biological matrices with analytes in low concentration or for drug seizure samples. Further work for the evaluation of analytical performance parameters and the validation of this promising coupled technique is underway to extend the applicability of the method.

## Data Availability

The data presented in this study are available on request from the corresponding author.

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
