# Peer review of "Rapid Determination of the ‘Legal Highs’ 4-MMC and 4-MEC by Spectroelectrochemistry: Simultaneous Cyclic Voltammetry and In Situ Surface-Enhanced Raman Spectroscopy"

_sensors, 2021, doi:10.3390/s22010295_

Round 1
Reviewer 1 Report
The authors developed a SERS and electrochemistry dual-readout platform for mephedrone (4-MMC) and 4-methylethcathinone (4-MEC) detection based on a AgNPs-modified commercial bimetallic glucose test strip. The dual-readout detection of the two drugs can be achieved by the oxidation peak shift in CV curves and the differences of SERS spectra. The dual-mode detection seems interesting and meaningful. However, the manuscript could not be recommended for publication before addressing the following major concerns.
- Introduction still needs more literatures to state the current status and technical characteristics of both SERS and EC to establish the correlation of the two technologies.
- Authors should provide an experimental setup for the electrochemical SERS platform.
- The schematic diagram was insufficient to illustrate the principle of the method.
- The reproducibility of the SERS substrate is one of significant properties for SERS sensing. However, there was no related data in the manuscript. In addition, the stability investigation of SERS performance under pH=1.8 is suggested to be explored.
- The selectivity and interference measurements should be supplemented in the manuscript to investigate the selectivity of the method.
- How authors identify and quantify the two molecules in a mixed 4-MMC and 4-MEC sample?
- Authors should discuss the advantages of the proposed method with appropriate references.

Author Response
The authors developed a SERS and electrochemistry dual-readout platform for mephedrone (4-MMC) and 4-methylethcathinone (4-MEC) detection based on a AgNPs-modified commercial bimetallic glucose test strip. The dual-readout detection of the two drugs can be achieved by the oxidation peak shift in CV curves and the differences of SERS spectra. The dual-mode detection seems interesting and meaningful. However, the manuscript could not be recommended for publication before addressing the following major concerns.
- Introduction still needs more literatures to state the current status and technical characteristics of both SERS and EC to establish the correlation of the two technologies.
We appreciate the suggestion. We added 10 new references with corresponding citations in the introduction. (From line 79 to 99). The following excerpts has been added and some paragraphs have been modified:
- “Our group has reported the behavior of Cu centers of enzymes using nanostructured Au as SERS substrates [28].”
- “During a combine EC-SERS experiment, the metal surface activation occurs in situ in a controlled and straightforward way. The voltammetric pretreatment at the electrode surface allowed an efficient and reproducible development of roughened nanostructures [29–31]. The advantage of this procedure is that it combines Raman spectroscopy with electrochemical analysis, thus offering both identification and quantification capabilities [32]. Several EC-SERS miniaturized sensors have been described in a recent review, covering quality control, diagnosis, biomedical, environmental and food safety fields [33]. A novel electrochemical roughening technique of commercial disposable electrodes of metallic surfaces such as gold and silver has been used to improve the reproducibility of the SERS substrate based on the deposition of nanostructures [34–37]. Additionally, screen-printed electrodes (SPE) have been recently reported as excellent substrates for SERS experiments [38–40].
Also, the last paragraphs have been changed to explain in a better way how the experiments were correlated to both involved technologies (lines 92-99):
- “This manuscript presents a time-resolved spectroelectrochemical method for detecting two synthetic cathinones -4 MMC and 4-MEC- using a gold SPE electrode as SERS substrate (AuSPE). These two cathinones (Scheme 1) are structurally similar, differing only by a CH2 group, which challenges their identification when they are present as mixtures in diluents or other drugs. Some of the advantages and limitations of the EC-SERS approach are described. The purpose of using the CV technique to generate onsite nanostructures and as a semi-quantitative tool is also discussed”.
- Authors should provide an experimental setup for the electrochemical SERS platform.
- The conditions for the spectroelectrochemical determinations are listed in section 2.4. The substrates used in this research are commercial products sold specifically for SERS analysis by Methrom-DropSense.
- Added in this section, the term “EC-SERS platform”, alongside screen-printed electrodes for clarity.
- The schematic diagram was insufficient to illustrate the principle of the method.
We have redesigned the initial version of the graphical abstract to highlight the principle of the method and summarizing our research work in a simpler way. Thanks for this important observation.
- The reproducibility of the SERS substrate is one of significant properties for SERS sensing. However, there was no related data in the manuscript. In addition, the stability investigation of SERS performance under pH=1.8 is suggested to be explored.
The reproducibility of the SERS substrate was measured. Table 1 shows the coefficients of variation for the two main bands of the EC-SERS spectra obtained for the drug-maltose mixtures in the electrolytic support at pH 1.8 (working analysis condition).
The following section was included in the experimental part for a better understanding (lines 239 to 243):
“2.7 Reproducibility of the EC-SERS substrate”. The following text was added:
- “Simulated seized samples of 4-MMC and 4-MEC cut with maltose in a ratio of solid drug to cutting agent 1:4 was tested by the simultaneous technique. The measurements were performed in triplicate on different electrode sensors to assess the Raman intensity of the two principal bands in the EC-SERS spectrum” (section 3.3).
Also, this text has been included in the section 2.4 (lines 211 to 212): “These electrodes were selected to evaluate the reproducibility of the SERS substrate.”
- The selectivity and interference measurements should be supplemented in the manuscript to investigate the selectivity of the method.
A new subtitle has been added in the discussion (Interference study, section 3.4, lines 526 to 529), and a brief paragraph: “The interference effect was evaluated by testing simulated seized drugs cut with maltose and lidocaine, very common cutting agents for this type of psychoactive substances [61].The selectivity has been discussed in a new paragraph (lines 577 to 584):
“The selectivity of the synchronous experiment can be evaluated independently. The electrochemical activity of the substances present in the mixture and the oxidation waves developed at the gold electrode caused limited selectivity on the target analyte peaks. Interestingly, when the EC-SERS was set, the selectivity improved. Several characteristic bands of the target molecules were enhanced and allowed the proper identification of each drug with contrasting responses when the mixture involved maltose and lidocaine. Adequate identification and quantification were possible in the seized drug cut with maltose only.”
- How authors identify and quantify the two molecules in a mixed 4-MMC and 4-MEC sample?
The quantification aspect is a possibility offered by the method. However, this topic was outside the scope of our research that focused mainly on identifying drugs. Although both drugs were analyzed mixed with cutting agents and diluents, it was not considered to analyze a sample doped with the two mixed analytes because it is not common to find these drugs mixed in a street seizure sample. A citation by Ayres and Bond [63] was added in the discussion. (Line 641). A hyperlink to this paper is shown below for easy access.
https://doi.org/10.1136/bmjopen-2012-000977.
- Authors should discuss the advantages of the proposed method with appropriate references.
The general discussion was supported with more references. Additionally, the following text was added to point out one of most important advantages of the proposed method: (Line 633 to 636):
- “An outstanding advantage of this method is the implementation of the SERS substrate electro-activation procedure for the quantification of the two drugs studied, either in samples mixed with their cutting agents or in biological matrices in which the analyte concentrations are very low”.
Reviewer 2 Report
The objective of this work by González-Hernández et al. is to develop a sensing method that can quantitatively characterize 4-MEC and 4-MMC in the same reaction mixture. Their work tackle the challenge of electrochemically distinguishing between 4-MMC and 4-MEC due to their very similar oxidation potentials as well as the challenge of teasing them out in other electrochemically active media. The experimental design enables in situ oxidation and reduction of Au electrode along with the analyte which results in SERS signal from the two analytes that differ in one Raman band. Overall I find their conclusions largely supported by their data, although the authors didn’t include how quantitative analysis can be applied to simulated samples. I therefore would recommend publication after the following comments are addressed.
(1) The authors claim peak II is the oxidation of Au whereas peak I is the oxidation of the substrate. But the oxidation peak of blank Au is in between peak I and peak II. Please briefly explain this in the text.
(2) The authors can try to use size distribution analysis for Figure 3A and 3B to support their claim on structural change of Au after electrochemical cycling.
(3) The examples listed in the article are more qualitative in nature than quantitative. The authors should supply information on how quantitative analysis are performed using this method, including the detection limit, the calibration curve of Raman peak area versus analyte concentration. Without this information, it’s hard to judge the true value of this methodology.
Author Response
The objective of this work by González-Hernández et al. is to develop a sensing method that can quantitatively characterize 4-MEC and 4-MMC in the same reaction mixture. Their work tackle the challenge of electrochemically distinguishing between 4-MMC and 4-MEC due to their very similar oxidation potentials as well as the challenge of teasing them out in other electrochemically active media. The experimental design enables in situ oxidation and reduction of Au electrode along with the analyte which results in SERS signal from the two analytes that differ in one Raman band. Overall I find their conclusions largely supported by their data, although the authors didn’t include how quantitative analysis can be applied to simulated samples. I therefore would recommend publication after the following comments are addressed.
- The authors claim peak II is the oxidation of Au whereas peak I is the oxidation of the substrate. But the oxidation peak of blank Au is in between peak I and peak II. Please briefly explain this in the text.
The following text has been added (Lines 335 to 340):
- “The electro-oxidation of both substances occurs at a potential of around +0.91 V (peak I) as suggested by the growth of the current peak when increasing the concentration from 50 mg/mL to 100 mg/mL, while gold oxidizes around +1.1 V (peak II). The prominent cathodic peak at +0.62 V for drug samples (peak III) or +0.52 V for the blank, corresponds to the reduction of the gold compounds previously formed in the positive scan. Both oxidation and reduction peaks are shifted towards less positive potentials for the blank”.
Furthermore, a shift towards less positive potentials is observed both for the gold oxidation peak and for the reduction (influence of the electrode history).
- The authors can try to use size distribution analysis for Figure 3A and 3B to support their claim on structural change of Au after electrochemical cycling.
This is a very valuable observation. We have performed the analysis of the SEM images with ImageJ and we have found results that support what we mentioned initially in the discussion.
The following procedure has been added to the experimental section 2.3 (Lines 202 to 205):
- “The SEM images were processed and analyzed using ImageJ. The scale was set to define the pixels in terms of SEM scale. A fast Fourier transformation (FFT) and a band-pass filter were used. The threshold was adjusted to analyze particles of size larger than 0.1 mm2”.
Additionally, the following statement es now included in the discussion (Lines 422 to 424):
- “The particle-size distribution analysis demonstrated an increase of 17% for nanoparticles with a size of 0.1 mm and 50% for nanoparticles with a size around 0.3 mm after voltammetric treatment”.
- The examples listed in the article are more qualitative in nature than quantitative. The authors should supply information on how quantitative analysis are performed using this method, including the detection limit, the calibration curve of Raman peak area versus analyte concentration. Without this information, it’s hard to judge the true value of this methodology.
The spectroelectrochemical method developed in this work effectively identified two synthetic cathinones as representatives of seized drug materials with minimum differences in their chemical structure (via the EC-SERS data). Additionally, the CV voltammetric method offered an alternative quantification during the electro activation procedure. The calibration curves generated were already included in the article. Performance parameters like linearity and limits of detections have been calculated and reported. (Lines 404 to 406):
- “The limit of detection (LOD) was estimated at three times the standard deviation of the linear regression divided by the slope of the linear curve (3σ/S). The method for 4-MMC exhibited a LOD of 6.6 mg/mL and 2.4 mg/mL for 4-MEC”.
For simulated samples we recommend a more sensitive technique such as DPV or SWV since the expected drug concentration in biological samples is in the order of parts per billion (ng/mL). CV is a technique that works best for higher concentrations such as seized drugs scenarios. Our research group is currently working in exploring more sensitive voltametric techniques as indicated in the conclusions: (Lines 654 to 656)
- “Further work for the evaluation of analytical performance parameters and the validation of this promising coupled technique is underway to extend the applicability of the method”.
Round 2
Reviewer 2 Report
I thank the authors for the revised manuscript. I can now recommend it for publication.